# Comparative Study of Different Acidic Surface Structures in Solid Catalysts Applied for the Isobutene Dimerization Reaction

**DOI:** 10.3390/nano10061235

**Published:** 2020-06-25

**Authors:** José M. Fernández-Morales, Eva Castillejos, Esther Asedegbega-Nieto, Ana Belén Dongil, Inmaculada Rodríguez-Ramos, Antonio Guerrero-Ruiz

**Affiliations:** 1Dpto. Química Inorgánica y Técnica, Facultad de Ciencias, UNED, c/Senda del Rey No. 9, 28040 Madrid, Spain; jmfernandez@ccia.uned.es (J.M.F.-M.); aguerrero@ccia.uned.es (A.G.-R.); 2Dpto. Ingeniería Química, Facultad de Ciencias, UCM, Avda. Complutense s/n, 28040 Madrid, Spain; 3Instituto de Catálisis y Petroleoquímica, CSIC, c/Marie Curie No. 2, Cantoblanco, 28049 Madrid, Spain; a.dongil@csic.es (A.B.D.); irodriguez@icp.csic.es (I.R.-R.)

**Keywords:** catalysts, dimerization, isobutene, olefins

## Abstract

Dimerization of isobutene (IBE) to C_8s_ olefins was evaluated over a range of solid acid catalysts of diverse nature, in a fixed bed reactor working in a continuous mode. All catalytic materials were studied in the title reaction performed between 50–250 °C, being the reaction feed a mixture of IBE/helium (4:1 molar ratio). In all materials, both conversion and selectivity increased with increasing reaction temperature and at 180 °C the best performance was recorded. Herein, we used thermogravimetry analysis (TGA) and temperature programmed desorption of adsorbed ammonia (NH_3_-TPD) for catalysts characterization. We place emphasis on the nature of acid sites that affect the catalytic performance. High selectivity to C_8s_ was achieved with all catalysts. Nicely, the catalyst with higher loading of Brønsted sites displayed brilliant catalytic performance in the course of the reaction (high IBE conversion). However, optimum selectivity towards C_8_ compounds led to low catalyst stability, this being attributed to the combined effect between the nature of acidic sites and structural characteristics of the catalytic materials used. Therefore, this study would foment more research in the optimization of the activity and the selectivity for IBE dimerization reactions.

## 1. Introduction

The energy dependence of fuels obtained from fossil sources continues to be a serious problem in many countries that do not count with such natural reserves. Without going any further, in 2017, gross imports of massive energy into the European Union (EU) stood at 87% [1], highlighting transport as the sector that consumes the most energy (33%) [2]. In addition, this sector is deeply dependent on fossil fuels, since 95% of the energy it uses is derived from these sources [3]. This energy dependency is further troubling when taking into consideration that we are currently running out of these sources and, in addition, the fuels that are extracted are the main contributors to climate change. Therefore, there is a need to progressively replace the non-renewable energy sources by inexhaustible ones. One of these renewable sources for fuel production is biomass [4], which is a sustainable carbon resource with neutral CO_2_ emissions.

In the last decade, important studies have been conducted with the aim of producing fuels, using butanol as an intermediate, which in turn is a byproduct of biomass. This compound can be dehydrated to butene and subsequently undergo oligomerization reactions to produce from C_8_ to C_16_, and after subsequent hydrogenation yield the desired fuels [5,6]. One of the most used hydrocarbons in the fuel market is isooctane, because it is a non-aromatic compound, has a high-octane rating, has a low sulfur content and low volatility [7]. This compound can be obtained from isobutene (IBE), which is a relevant molecule employed in the synthesis of polymers, as well as gasoline additives such as methyltertbutyl ether (MTBE), by reaction with an alcohol [8], or isooctane itself. Industrially, isooctane can be obtained from the alkylation reaction of IBE in liquid phase with strong acids such as HF or H_2_SO_4_. However, the use of these acids as homogeneous catalysts make their regeneration very expensive, apart from the environmental problems that they can provoke due to their corrosive capacity and water solubility [9,10]. Therefore, for years, an alternative route has been studied with the use of solid acid catalysts, which make safer handling possible and provide a lower environmental impact. In the literature among the solid acid catalysts used for this reaction we find ion exchange resins [6,11,12,13], sulfated metal oxides [14,15,16,17,18], heteropolyacids [19,20,21], zeolites [22,23,24,25,26] or Metal-Organic Frameworks (MOFs) [27,28].

The studied reaction in this work consists on IBE to produce isooctane C8 such as 2,4,4-trimethylpen-1-ene (TP1) and 2,4,4-trimethylpen-2-ene (TP2). This alkylation reaction of IBE needs an acidic medium or a metal capable of breaking the double bond, which generates a tertiary carbocation that enables interaction with another IBE molecule present in the medium. It is an acknowledged fact that the surface acidity of the catalyst plays a relevant role on the IBE dimerization reaction. However, the acidity or the presence of metal can also promote chain polymerization reactions, which could be minimized working at low conversions, between 20% and 60%, or decreasing the contact time with the catalytic bed. Thus, catalysis systems achieving high catalytic activities and optimum selectivity towards dimers in the oligomerization of IBE still remains a challenge.

In this reaction we have used the following catalysts: an ion exchange resin (Amberlyst 15), a carbon supported heteropolyacid, a sulfated zirconium oxide and a NiO/Al_2_O_3_ catalyst. The common denominator of these catalysts is their surface acidity, since in the structure of the Amberlyst we can find sulfonic groups (-SO_3_H) [6], in that of the heteropolyacids there is the presence of protons attached to the metal cation that makes them considered superacids [21] and in the sulfated zirconium oxide the presence of sulfate groups bound to the oxide structure [15,16,17,18]. On the other hand, the incorporation of NiO into the alumina structure was consistent with the fact that the metal atom played the role of reaction intermediate, being able to interact with the electrons of the double bond in the IBE molecule; thereby allowing the formation of the carbocation and the attack of a new IBE molecule for alkylation to occur [29,30,31].

A competent method is pursued in order to orientate the catalytic behavior in reactions based on solid acid-catalysts. From another perspective, this research would decipher the relevant role of adequate textual properties of the materials required in these particular reactions. A great deal of effort is undertaken in order to reveal the nature of active site which is responsible for IBE dimerization. All these tests employing different catalysts have the objective of accomplishing the IBE dimerization reaction selectively and, therefore, avoiding polymerization reactions that generate carbonaceous species which would lead to the deactivation of the catalyst.

On the other hand, another motivation for this study with solid acid catalysts is that most of the research on oligomerization of butenes is performed in liquid phase at moderately low temperatures, though, high pressure is required. Therefore, additional specialized components are needed to cautiously carry out the reaction. Contrary to this, in our work, these requirements are ruled out as IBE dimerization reaction is carried out under atmospheric pressure conditions in a gas-phase continuous flow reactor. 

## 2. Materials and Methods 

### 2.1. Catalysts Preparation

The Amberlyst 15 catalyst (denoted A15) was supplied by Alfa-Aesar, Kandel, Germany (S_BET_ = 15 m^2^/g, CAS: 39389-20-3). Amberlyst 15 presents high acidity due to its content of sulfonic acid groups, equal to 4.7 mmol/g (https://www.sigmaaldrich.com/catalog/product/sial/06423?lang=es&region=ES). Before its use, the catalyst A15 was purified. The commercial resin beads (0.5 mm) were incessantly washed with acetone, dried at 110 °C during 24 h and kept in a desiccator. Prior to all catalytic tests, the grains were loaded into the reactor, which was thereafter heated at 180 °C for 2 h under helium flow. 

The second catalyst, denoted STA/HSAG_100_, was a heteropolyacid or polyoxometalates supported on graphite HSAG_100_. STA (H_4_SiW_12_O_40_·nH_2_O, CAS: 12027-43-9) was supplied by Sigma-Aldrich (Darmstadt, Germany). Commercial high surface area graphite HSAG_100_ supplied by TIMCAL (Bodio, Switzerland, CAS: 7782-42-5) with a specific surface area of 100 m^2^/g (particle size <125 μm) was chosen as the support for STA. This catalyst containing an STA loading of 15 wt.%, was synthesized by the incipient impregnation of HSAG_100_, employing as solvent for the STA solid a mixture of ethanol/water (1:1 ratio) as reported previously [32]. Before each experiment, STA/HSAG_100_ was inserted into the reactor and heated prior to the reaction at 180 °C for 2 h under a helium flow of 20 mL/min.

Commercial ZrO_2_/S Zirconium Hydroxide (Sulfate doped, CAS: 34806-73-0) Grade XZ0682/01 was purchased from Mel (Darmstadt, Germany). ZrO_2_/S Zirconium Hydroxide precursor was calcined in flowing air at 600 °C for 5 h to obtain active sulfated zirconia (ZS) with a specific surface area of 116 m^2^/g [33]. This catalyst was also treated prior to its use in a helium flow of 20 mL/min at 180 °C. 

NiO/Al_2_O_3_ catalyst was prepared by incipient impregnation of γ-Al_2_O_3_ (CK-300 from Cyanamid Ketjen, S_BET_ = 180 m^2^/g, CAS: 1344-28-1) with an aqueous solution of NiSO_4_·6H_2_O (also provided by Sigma-Aldrich, CAS: 10101-97-0). The theoretical metal (Ni) amount incorporated was 5% W/W. Subsequently, the catalyst was dried at 100 °C overnight and calcined at 500 °C during 5 h. The final sample, named NiO/Al_2_O_3_, was also treated prior to its use under a 20 mL/min helium flow at 180 °C. This catalyst was also treated in situ with pure hydrogen flow in order to reduce the Ni ions to their metallic state. 

### 2.2. Supports and Catalysts Characterization

All the samples were characterized by thermogravimetric analyses (TGA). TGA were conducted under helium using TA Instruments (New Castle, DE, USA), model SDT Q600 TA System. About 5 mg of the sample was deposited in an alumina crucible and the temperature was raised until to 800 °C (starting from room temperature) at a rate of 10 °C/min. In order to remove H_2_O, on reaching 100 °C, the temperature was kept constant for 30 min. The measured weight loss gives information on the changes in the solid sample composition and thermal stability features. A derivative weight loss curve (DTG) can be used to determine the temperature at which the rates of weight changes are maximum. Gases issued at the outlet of the TGA equipment are CO_2_, CO, H_2_, O_2_ and SO_2._ This was evidenced by a Mass Quadrupole Spectrometer (Pfeiffer Vacuum Omnistar^™^ GSD 301, Asslar, Germany) coupled to the TGA, this is the technique abbreviated as temperature programmed desorption (TPD)-MS. On the other hand, total acidity of the samples was estimated with the aid of static ammonia adsorption using an ASAP 2010 Micromeritics unit (Norcross, GA, USA). The adsorption was carried out at 50 °C and 50 mg of material were analyzed. The samples were first pretreated in helium at 230 °C (5 °C min^−1^) for 60 min after which it was cooled to 50 °C and then NH_3_ was adsorbed during 30 min. After flushing with He the amount of desorbed NH_3_ was determined ramping (10 °C/min) from 50 °C until 230 °C (NH_3_-TPD). This final temperature was maintained for 30 min and from the analysis of the gas evolved by a thermal conductivity detector (TA Instruments, New Castle, DE, USA) the amount of chemisorbed ammonia can be detected. 

### 2.3. Catalytic Evaluation in Gas Phase Isobutene Dimerization 

The IBE dimerization reaction was carried out in a continuous flow fixed-bed tubular reactor contained within a PID Eng&Tech system (Alcobendas, Spain), equipped with a furnace using PID control and counting on mass flow controllers mod. EL-Flow Select (Bronkhorst High-Tech B.V., Ruurlo, The Netherlands). Before reaction, the catalyst (50 mg) was pre-treated under He atmosphere (20 mL/min) at 180 °C for 2 h. Any substantial mass-transfer limitation was hindered applying high linear velocity and the adequate catalysts grain size (sieve fraction 0.25−0.5 mm). Thereafter, the system was cooled and the catalytic tests were carried out feeding the reactor with IBE:He in a 1:4 molar ratio counting with a total gas flow of 8 cm^3^/min^−1^. The catalytic measurements were done once the catalyst was brought to the required reaction temperature (180 °C) under He. Thereafter, the gas stream was switched to the reaction mixture. Conversion measurements were conducted every 10 min after stabilizing the samples. The IBE conversions were derived upon the on-line analysis of the exit gas with the aid of a Varian CP-3800 chromatograph (Agilent, Santa Clara, CA, USA) equipped with a FID detector (200 °C, detection range 10) and a Supelco alumina sulfate PLOT capillary column (30 m × 0′53 mm/10 μm). The programmed temperature of the column is as follows: it was maintained at 120 °C during 5 min; then a raised until 180 °C at a rate of 15 °C min^−1^ and kept constant at 180 °C for 10 min. IBE conversions, activity and selectivity were estimated as:Conv IBE (%) = (([IBE]° − [IBE])/[IBE]°) × 100(1)
where Conv IBE (%) (1) is IBE conversion percentage, [IBE]° and [IBE] are the inlet and outlet IBE concentrations, respectively.
Selec TP1 (%) = (2 × [TP1])/(2 × Σ_i_ [products]_i_) × 100(2)
where Selec TP1 (%) (2) is percentage of 2,4,4-trimethylpent-1-ene produced, [TP1] represents the concentration of TP1 produced and Σ_i_ [products]_i_ is total concentration of products.
Selec TP2 (%) = (2 × [TP2]_i_)/(2 × Σ_i_ [products]_i_) × 100(3)
where Selec TP2 (%) (3) is percentage of 2,4,4-trimethylpent-2-ene produced, [TP2] represent the concentration of TP2 produced and Σ_i_ [products]_i_ is total concentration of products.
Selec C8s (%) = (2 × [C_8_s]_i_)/(2 × Σ_i_ [products]_i_) × 100(4)
where Selec C8s (%) (4) is percentage of another C8_s_ produced, [C_8s_] represent the concentration of C_8s_ produced and Σ_i_ [products]_i_ is total concentration of products.
Activity (μmol IBE converted/g_cat_) = (Conv IBE (%) × [IBE]°/g_cat_)(5)
where Activity (5) is catalytic activity, Conv IBE is IBE conversion percentage, [IBE]° are inlet IBE concentration and g_cat_ is mass of catalysts.

In order to verify that the solids exhibited a reproducible behavior the catalytic tests were carried out in duplicate.

## 3. Results

### 3.1. Catalysts Characterization

The acidity of catalysts was studied by NH_3_-TPD and their profiles are shown in Figure 1. As it is well-known, the temperature at which NH_3_ desorbs can be related to the strength of acid sites. However, NH_3_ is a strong non-site-specific base that can be adsorbed on both Brønsted and Lewis acid sites, and simultaneous desorption from several sites can occur [34]. The experimental conditions employed in the present work allow us to disregard the contribution of gaseous or adsorbed NH_3_ as well as those from species evolved from the material itself and to consider, exclusively, contributions ascribed to acid sites. In addition, under these low temperatures, it is unlikely that surface reduction by ammonia and, hence, change in the acidic properties might take place. Moreover, none of the catalysts hold micropores, so ammonia desorption temperatures should not be influenced by diffusion constrains.

Overall, it can be observed (Figure 1) that the temperature range in which NH_3_ desorption occurs is similar for all the samples, and that the maximum is shifted to higher temperatures following the order: NiO/Al_2_O_3_ > ZS > A15 > STA/HSAG_100_. Moreover, the intensity of the STA/HSAG_100_ TPD-NH_3_ profile is significantly lower compared with that of the other samples. This was expected since HSAG_100_ is a graphitic support that barely holds any acid site; hence the acidity of this catalyst is given mostly by the STA which is in a 15 wt.%. However, it is worth highlighting that the heteropolyacid also retains its acidity despite being supported, and this suggests a weak interaction between STA and the graphite as observed previously in silica-supported systems [35]. In contrast, the other tested catalysts are either bulk catalysts, A15 and ZS, or NiO 5 wt.% supported on alumina, so the total acidity is obviously higher as it emerges from the structure and the surface groups themselves. 

In order to analyze the results, the NH_3_-TPD profiles were fitted by two symmetrical peaks. The resulting peaks, with maxima in the range of 174–187 °C and 224–246 °C, represent sites of increasing acidity strength. The first contribution is observed at quite similar maximum temperature for all the samples except for A15 that displays the maximum at higher temperature, ca. 187 °C. The second contribution is observed at the same temperature for NiO/Al_2_O_3_ and A15, ca. 241 °C, while it is slightly shifted to higher temperature for STA/HSAG_100_, ca. 246 °C and to significantly lower temperature for the sample ZS, ca. 224 °C. 

The ratio of these sites LT/HT (Low T: LT, and High T: HT) was also estimated (in Figure 2), and results indicate that it follows the trend ZS > NiO/Al_2_O_3_ > A15 > STA/HSAG_100_. These values indicate that, even though ZS presented less acid sites in the HT range, as refers to the desorption temperature, its LT/HT ratio is the highest among the samples, i.e., 1.21. In contrast, both A15 and STA/HASG_100_, despite having stronger acid sites than ZS as evidenced in this higher temperature of desorption, hold significantly lower LT/HT ratios, i.e., 0.64 and 0.57, respectively. 

We have evaluated four samples with acid sites of different nature, both Brønsted (mainly SO_3_H, H^+^ and -OH from the support surface) and Lewis (Ni, Zr, Al), all of them contributing to weak and strong acid sites [36]. The evaluated samples have surface -OH groups that could be responsible for the acidity given by the first desorption peak. The shift to higher temperatures of the LT peak observed in A15 could be tentatively ascribed to the contribution of the H^+^ of this cation exchange resin, which would be more acidic than the -OH groups. The second contribution could also include the desorption of NH_3_ from -OH groups along with desorption from more acidic sites like Lewis, e.g., Al and Ni^+δ^, as observed previously [37,38]. The contribution derived from –SO_3_H groups of the ZS and A15, which correspond to Brønsted acid sites, have been reported to occur at higher temperatures [39]. 

Table 1 and Figure 3 summarizes the total weight loss at 600 °C for the studied catalysts. The TGA technique offers information on the stability of the catalysts as refers to the loss of water and their subsequent thermal decomposition. Generally, the catalysts present a low weight loss being stable at the reaction temperature range. It should be noted that the catalysts had been pretreated prior to the reaction at 180 °C for several hours, assuring the removal of water species in all cases. The analysis of TPD-MS showed the evolution of m/z 64 ascribed to desorption of SO_2_ on ZS, between 350–390 °C associated with sulfonic acids groups. While STA/HSAG_100_ and NiO/Al_2_O_3_ reveal a weight loss due to decomposition at higher temperatures (>350 °C). Finally, the TGA of A15 has been extensively reported [40,41] revealing an intensive loss of water at temperatures near 200 °C. The corresponding TGA and DTG curves are presented in Figure 3. The mass loss observed within 100 °C is associated with the removal of physically adsorbed molecular water. However, the significant loss of water at temperatures near 200 °C of A15, leads to an unclear comparison among samples. Therefore, the exact values are presented in the Table 1 that can be taken into account for comparison purposes.

### 3.2. Catalytic Results

After the analysis of the precedent information, it seemed appropriate to investigate the dimerization of IBE over catalytic materials that possess these properties. IBE oligomerization is representative of acid-catalyzed reactions where the main defiance involves procuring high selectivity to C_8_ (Scheme 1), inhibiting the formation of heavier olefinic products (C_12_ and C_16+_) and improving the catalyst lifespan at the same time.

Figure 4 depicts the catalyst activity (5) results, over the different samples at 180 °C with respect to time on stream. As portrayed, the highest initial activity for IBE transformation was found with A15 and STA/HSAG_100_ catalysts. However, a noteworthy diminution of this parameter within the first four hours was observed (about 60–75%). It is interesting to notice the catalyst deactivation with time on stream, especially at the early stages of the reaction (rapid initial deactivation). Thereafter, the catalysts’ activities were maintained constant for another 24 h of the process (not shown). The IBE activity was lower for ZS and NiO/Al_2_O_3_ than that obtained with A15 and STA/HSAG_100_. 

Figure 5 describes the results which emphasize the influence of the reaction time on the selectivity towards C_8_ olefins, produced with the aid of the different catalysts. The on-stream analysis of reaction products evidenced that C8s, mainly TP2 (3) and TP1 (2), were the only products obtained. High molecular weight oligomers were not identified at any of the studied temperatures. Nonetheless, compounds of the sort are surely formed during the process, though may not be accounted for as they remain confined at the catalysts’ surface (see the discussion on the catalyst deactivation). STA/HSAG_100_, ZS and NiO/Al_2_O_3_ presented high selectivity to TP2 (3) (around 70%) being the remaining 25% to TP1 (2). It should be stressed that the IBE conversion to TP2 offered the lowest values in the case of A15, see Figure 5a, and the formation of other C8s olefins (4) (not reported in Figure 5) was significant higher (20% at 150 min) compared with STA/HSAG_100_, ZS and NiAl_2_O_3_ (less 10%). Overall, all catalysts exhibited high activity toward the selective production of TP2 (3) olefin. The high selectivity efficiency of the catalysts in the IBE dimerization disclosed in this work is of immense relevance because, as far as we can tell, such high selectivities to C_8_ products have not been divulged, so far, by other researchers under continuous flow reaction conditions.

The effect of reaction temperature on IBE conversion (1) has been previously studied within the 50–180 °C temperature interval (see Appendix A). The reason behind this is, in the first place, concerning the fact that this oligomerization reaction is highly exothermic [42,43]. The temperature effect on IBE dimerization conversion as well as in subsequent oligomerizations, was considered in order to maximize the amount of C8s obtained in the reaction. The chosen studied temperatures were selected after numerous preliminary experiments. In addition, some tests were conducted in the absence of catalyst, seeking the reactant’s activity at high reaction temperatures. In addition, employing the same reaction conditions, blank experiments of IBE reaction with just the bare supports (HSAG_100_ and Al_2_O_3_) and pure STA were carried out in order to check their inherent catalytic activity. Furthermore, external mass transfer resistance and pore diffusion are found to be non-limiting factors in this catalytic application of these materials.

## 4. Discussion

The IBE dimerization is a complex, extremely exothermic process whereupon a succession of consecutive and parallel reactions unfolds. Due to this, without neglecting the formation of high molecular weight derivatives, such as C_12_ and C_16_ olefins or even higher (C_20+_), the selectivity to a specific C_8_ olefins is the key parameter. 

Overall, the literature described reaction processes of IBE dimerizations via carbenium ion mechanism. As an example, Schmidl [44] explains how this reaction advances through this mentioned path (carbenium ion way). The process conditions and the relative stability of the intermediate carbenium ions will determine the reaction equilibria and hence the products obtained. The catalytic cycle of the solid acid isomerization includes three steps: 1—chain initiation to form the first active carbenium ions, 2—rearrangement of the carbenium ions and 3—chain propagation. Solid acid catalysts release a proton, which favors alkene protonation to form an active carbenium ion. The dimerization of light olefins, such as IBE, in the solid acid catalyst presence, is based on the consecutive reaction sequence going on through carbocation intermediates. The addition of a proton (from the solid acid catalyst) to an olefin leads to a t-butyl cation formation, which then combines with another C_4_ to give the corresponding carbocation with eight atoms of carbon. This C_8_ may isomerize via hydride transfer and methyl shifts to form more stable cations. The presence of suitable amounts of Brønsted and Lewis acid sites is ascribed to favorable activities and selectivities. However, it is a current challenge to relate acid (Lewis and Brønsted) sites with activity or selectivity. 

The resin material Amberlyst “A15” used as catalyst in this study has been selected due to its structural and chemical complexity. It consists of a styrene-divinylbenzene-based support where the active sites are sulfonic groups. Hence, and as it was detected earlier on (see Section 3.1) by ammonia adsorption, the surface acid capacity is provided by sulfonic acid groups -SO_3_H and the distance between these acid sites. SO_3_ with an adjacent Brønsted acid site (H^+^) is regarded as the active site. In fact, Xiaolong Zhou et al. [12] reported that the surface acidity capacity of A15 is about 4.60 mmol H^+^/g. These acid sites of A15 determine the IBE conversion (1) and dimerization selectivity, where the density of sites or distance between them plays a decisive role. 

It can be deduced on viewing Table 2 that the initial conversion of IBE (1) (for A15) was near 50% and that the principal dimer (close to 60%, w/w, Figure 5a) is TP2. It can be derived from the data given in Table 2 that IBE conversion (1) is almost quantitative for weight–time values reaching 300 min.

As was already mentioned, initial high conversion was obtained over commercial Amberlyst A15. It is evident that IBE conversion (1) rises with increasing of active sites and with the optimal distance between acid sites. Since A15 presents mostly sulfonic acid groups -SO_3_H, it would seem logical that the catalytic results are mainly due to these Brønsted acidic sites. However, especially at the beginning of the process, the catalyst activity rapidly decreased. The conversion of IBE over A15 gradually diminished achieving a ~90% of conversion decrease, making obvious the deactivation behavior. Apparently, a higher amount of Brønsted sites or a not suitable enough distance led to the fast-initial deactivation of the catalyst. When the number of the Brønsted sites drop or the distance between them increases, the concentration of IBE adsorbed on the catalyst would decrease to a certain extent that could be beneficial to the dimerization process. In addition, the selectivity to a particular C_8_ could increase (mainly TP2) while that towards other C_8s_ decreases.

The principal cause for the catalyst deactivation seems to be that the surface of the solid acids is not capable of releasing C_8_ products formed at the catalyst surface rapidly enough and at the same time, they favor the augment of side reactions. In this sense, the acid sites are especially active for oligomerization causing side reactions. These side species serve as starting points for the growth of carbonaceous species that ultimately masks the active sites making them inaccessible and leading to deactivation [19]. Corma and Ortega [45] also highlight that reactants’ and products’ adsorptions on the active sites could play a vital role in catalyst deactivation. In order to avert these limitations, most industrial processes for IBE dimerization function at conversion values between 20% and 60% [43]. The low thermal stability of sulfonic groups and their decomposition under the conditions utilized in this study could be another reason for the catalyst deactivation. However, TG profiles indicate that the sulfonic group stability is quite high under the conditions elected this study. In addition, some reaction tests involving the incorporation of steam in the reactor feed show that deactivation is not related with the loss of structural water from the A15 material, since the catalyst also ended up deactivated. 

Catalytic properties will be compared with those of Amberlyst “A15”. In the first place, with the purpose of clarifying the effects of surface acidity, we have compared Amberlyst (A15) and H_4_[SiW_12_O_40_] (STA) supported over a high surface area graphite HSAG_100_ (STA/HSAG_100_) (Figure 4). Pure heteropolyacids usually display very low catalytic reactivity due to their reduced surface area. As aforementioned we had verified that HSAG did not exhibit significant activity for IBE conversion. Therefore, supported heteropoly acids, commonly prepared by their impregnation over classical high surface area materials, are adequate for catalytic reactions. Moreover, their strong acidity makes them attractive candidates to tackle the current challenges found in alkene dimerization.

In ours experiments of IBE conversion and product yield (Figure 4 and Figure 5), the evolution with the reaction time of catalytic parameters can be compared for both catalysts (A15 and STA/HSAG_100_). Both catalysts presented an initial high activity in the IBE dimerization gas phase reaction. The acidity of these materials was enough to activate this reaction, observing TP1 and TP2 as the main reaction products. However, one main aspect can be featured: A15 is initially (during the first 50 min) more active than STA/HSAG_100_, the latter becoming less deactivated than A15 upon the first 50 min into the reaction. Deconvolution of TPD-NH_3_ portrayed similar profiles at low temperature (LT) and high temperature (HT), being LT/HT similar in both samples. As refers to the acid sites, Guerrero et at. [46] studied the surface acidity of STA/HSAG in the decomposition reaction of isopropanol. The sole product obtained in this study was propylene, which results as a dehydration product favored by the presence of acidic surface centers. Therefore, from the estimation of specific activities for isopropanol, they could achieve a direct and quantitative estimate of acid sites, this being higher for STA. From these results, it was obvious that the density of acid surface sites exposed on A15 material is somewhat lower than on the STA supported sample. This subtle difference cannot justify the lower initial catalytic activity of STA/HSAG_100_. Hence, this commercial material (A15) probably possesses acid surface sites not similar in nature or of different distances between neighbor surrounding acid sites, when compared with the acid sites of STA/HSAG_100_. 

Concerning the STA/HSAG_100_ catalyst, the graphite support presents a basic nature even though it is not functionalized with any surface specie. This is due to the delocalized p-electrons present on the basal planes of carbon supports [47]. On the other hand, the heteropolyacid solids anchored could be accommodated on defective carbon surface sites or at the graphite crystallite edges, which will be impeded by the oxygen groups (of acidic nature), whose exit creates defective sites. A good dispersion of the STA on the graphite could take place by an effective interaction of STA with the basal graphitic surfaces, i.e., small crystallites sizes. Furthermore, carbon supports can facilitate the products’ diffusion from the catalyst surface improving the IBE isomerization. Guerrero et al. [45] studied how for the graphite supported samples the heteropolyacid crystallites are practically undistinguished from the pristine support. It is worth emphasizing that the heteropolyacid also retains its acidity despite being supported as it was determined by NH_3_ adsorption. These compounds are highly acidic when the giver is a proton (Brønsted acidic sites) and they are sometimes considered as superacidic. It could be suggested that the combination of the particularly small STA supported crystallites, nature of the acid sites (Brønsted) and the surrounding atomic layout defines the activity of the catalysts. This cooperative effect leads to C8_s_ (mainly TP2) production. 

Finally, it should be noted that the STA/HSAG_100_ also suffers deactivation as a result of carbonaceous deposition. Furthermore, the comparison between both catalysts evidenced that the deactivation balance is worse in the case of A15 due to higher carbon deposits accumulation during process of the reaction on this material. In addition, STA/HSAG_100_ displays higher selectivity to C_8_ and higher catalytic durability than that of A15 in IBE dimerization. This can be attributed to the fact that the surface-appropriate acidity of former may be more effective in decreasing the interaction between the catalyst surface and coke precursors and this favors the readily desorption of coke deposits. This behavior also gives away some differences in the chemical and physical characteristics of the active sites displayed in STA/HSAG_100_ with respect to those in A15. Acid supported STA and A15 have proven to be active although they deactivate rapidly. This behavior is caused by formation of polyolefins as a result of the polymerization of IBE, leading to strongly adsorbed molecules on the acid sites, especially resin material. Therefore, it is necessary to prevent this poisoning of acid sites with an appropriate design of the catalyst.

Subsequently, the above mention catalysts were compared with the sulfated zirconia (ZS) sample. Activity versus time on reaction is reported in Figure 4, which reveals a rapid decline in the activity at the beginning of the catalytic run ascribed to a partial process of deactivation. This process of deactivation has already been described for A15 and for STA/HSAG_100_. After about 50 min, the activity stabilizes at values close to 15–18% remaining practically constant for at least 300 min. Note that the activity of ZS is lower than that of A15 and STA/HSAG_100_. However, its deactivation is less pronounced in comparison with that of A15 or STA/HSAG_100_.

Despite the fact that no a general agreement on the structure of ZS has not been achieved, it has been well established that acidic SO_3_ with an adjacent Brønsted site is the acidic active site [48]. Therefore, sulfated metal oxides such as ZrO_2_, have both Brønsted acid sites arising from the existing sulfate groups at the surface of the catalyst and Lewis acid sites derived from the metal oxides [49]. In our particular case of the ZS material, Lewis acidity could be inherent to Zr^4+^ sites. Both Brønsted and Lewis acid sites seem to be present on the ZS surface (Scheme 2). 

Xuan Menga et at. [50] studied the surface of sulfated zirconia synthetized by sulfated zirconium hydroxides. Here, Lewis acid are the main active sites on the surface, dependent upon the calcination temperature. Protonic (Brønsted) acid sites play a pivotal role, though the existence of plenty Lewis sites (made superacidic by the surface sulfates´ existence and inductive effect therefrom) could be crucial for the development of the IBE dimerization under the limitations applied in this study. Certainly, according to the results obtained with A15, high density and strength of Brønsted sites leads to a rapid deactivation by blockage of these sites. Therefore, there must be a correlation between strong acid sites and activity [51]. ZS catalyst present both Brønsted and Lewis sites on surface but the total acid centers of the pair decreased, which is reflected in the lower activity and yet a low deactivation rate in comparison with A15 or STA/HSAG_100_. Hence, it could be assumed that the acid sites (Brønsted and Lewis) must be in the accurate density and formation in order to carry out the reaction. The IBE isomerization could be thought to proceed over a Brønsted acid site with the Lewis acid site playing a supporting role. On the other hand, since the activity of ZS is low, its deactivation is also low. Now, the surface of the solid is capable of eliminating C_8_ products from the catalyst surface. As regards the deactivation, TGA and TPD-MS experiments have shown that low thermal stability of sulfonic groups or their decomposition under the limitations imposed in this study can be discarded. 

In contrast, NiO/Al_2_O_3_ catalysts showed a remarkably low catalytic activity, compared with the all the other catalysts (Figure 4), which reflected the ineffective strong acidity of this material. It should be recalled that blank experiments involving IBE and the bare supports (Al_2_O_3_) showed no catalytic activity. The γ-alumina acidic properties are well-known and usually ascribed to incompletely coordinated aluminum ions of Lewis acid nature [52]. Thermally activated alumina did not display strong Brønsted acid sites on its surface. Regarding the acidity of our catalyst (NiO/Al_2_O_3_), which was prepared by impregnation employing nickel (II) sulfate, it could exert certain influence on this property. The sulfate ions at the surface led to the occurrence of strong Lewis acid sites, if during calcination there is no decomposition of sulfate anion (SO_4_^2-^). The introduction of a certain amount of sulfur compounds to γ-alumina (verified by TPD-MS) is recognized to improve its acidity. The SO_4_^2-^ ions exist as a chelating bidentate ligand and enrich the Lewis metal centers’ acidity (Al and/or Ni) and the catalyst would be highly active for the oligomerization of olefins [53]. On the other hand, the Ni (II) sulfate impregnation on γ-Al_2_O_3_ and the subsequent calcination at 500 °C could provoke to the formation of either nickel aluminate or nickel oxide. The existence of only one type of surface nickel species should not be discarded in the calcined catalyst. In terms of activity, the catalyst treatment prior to its use under hydrogen flow has indicated that metallic Ni is not active under this reaction conditions. 

Normally, unsaturated Ni species interact with IBE in the course of the oligomerization reaction [53]. The surface species interact with the Ni close by, producing a strong acidity, which allows the activation of the dimerization of IBE with demand for strong acid sites. 

However, NiO/Al_2_O_3_ shows low catalytic activity most likely due to the insufficient acidity at the catalyst surface. The addition of NiSO_4_ as an acid precursor and the strength derived owing to this group has not induced the presence of strong Lewis acid sites and it has not permitted the activation of this demanding reaction. It should be emphasized that NiO/Al_2_O_3_ presents weak Lewis acid sites and that all the above discussed experimental evidences confirmed that the Brønsted-type acidity seem to be a more effective criterion in the IBE dimerization.

On the basis of the results obtained, it can be hinted that a high acid site concentration, type of acidity (Brønsted or Lewis) or distance among acidic sites must be the factors which allow the activation in the gas phase reaction. The thermal properties of these acid sites, as well as the textural properties also seem to be important. Moreover, the active adsorbed intermediated should be closely linked to the surface Brønsted/Lewis groups. The higher the acidity of the Brønsted sites, the higher is the reaction rate, which is in agreement with the following order of the total Brønsted acidity of the catalysts: A15 > STA/HSAG_100_ > ZS > NiO/Al_2_O_3_. However, it is worth noting that the correlation between the catalyst activity and the acid loading was not linear. This may suggest that in the process of IBE dimerization, not only surface chemistry (acid loading), but also textural properties of catalysts play a significant role.

It should also be mentioned that the selectivity (or products distribution) obtained with A15 was not the same as when using STA/HSAG_100_, ZS or NiO/Al_2_O_3_ where the main product was TP2. The results obtained clearly suggest that C_8_ olefins formation in the reaction must be rapidly and easily detached from the active sites, which would avoid the formation of C_12_ and C_16+_ olefin fractions and therefore the deactivating process. 

## 5. Conclusions

In this paper, the study of the IBE dimerization, under gas phase at moderately low temperatures, over different solid catalysts has led to outstanding results. The acidic sites present and their strengths have been associated with the catalytic activities and the respective selectivities to C_8_. These solids have proven to be highly selectivity to C8 (TP2 and TP1), disclosing the interest of this work. Moreover, the formation of minor amounts of others C_8s_ has been related with the existence of strong Brønsted sites. As a tentative explication, the solid acid catalysts release proton species that interact with close by active sites, producing a strong acidity which allows the activation of a gas phase reaction with demand for strong acid sites. In the case of Lewis sites, these seem to be less active than Brønsted sites. The catalytic performance of our catalysts might be assigned to a combination of their acidic and textural properties. Moreover, other factors such as high concentration of acid sites, thermal properties of these sites and ratio between Brønsted/Lewis sites or distance among acidic sites stimulate this reaction. On the other hand, if the acid sites are very active for the IBE dimerization, they would also favor the side polymerization reactions, modifying the surface acid capacity of the catalyst, which leads to rapid deactivation. In order to avoid limitations caused by deactivation for IBE dimerization, it is necessary to operate at relatively low conversions (20–60%) which will extend the lifespan of the catalysts. It is beneficial to prolong the durability of the catalyst. A positive effect on the catalyst stability could also be the use of an inert diluent in the IBE feed (in our case helium). This gas could be responsible for the “washing-out” of olefins stored at the active sites of the catalysts.

The high selectivity efficiency of the samples in the dimerization of IBE described in this study is of great importance as very high values (close to 100%) towards C_8_ fraction have been reported in this work. The dimers selectivity could be significantly enhanced by effectively controlling the acid capacity, among others. However, further work is still needed in order to know the ambiguities of the surface of the acid solids.

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
