# Peer review of "Comparative Study of Different Acidic Surface Structures in Solid Catalysts Applied for the Isobutene Dimerization Reaction"

_nanomaterials, 2020, doi:10.3390/nano10061235_

Round 1

Reviewer 1 Report

The paper could be published after the following minor corrections:

  • L237: "Table 2" should be "Table 1";
  • L489: "Bronsted" should be "Brønsted".

Author Response

Comments:
The paper could be published after the following minor corrections:
• L237: "Table 2" should be "Table 1";
• L489: "Bronsted" should be "Brønsted".

Response: We appreciate the comment of the referee and all typing errors have been corrected according to the referee’s suggestions

Reviewer 2 Report

This paper describes a comparative study of four different acidic catalysts, such as, Amberlyst 15, heteropolyacids on graphite, sulfated zirconia, and NiO-Al2O3, on dimerization of isobutene into C8 olefins. The authors thoroughly studied a nature of acidic sites (Bronsted and Lewis) in these catalysts. They found that the the most effectively dimerization took place on catalysts possesing Bronsted acid sites. However, they stated that isobutene dimarization is not only surface process on acidic sites of a  catalyst, but it can be very much dependable on textural properties of a catalyst. It was foubd that Amberlyst is the most effective catalyst for this dimerization.

In my opinion, the text of the paper is too long. Sometimes, it is difficult to follow the text, especially, when a lot of abbreviation are used in the text. My recomendation is to reduce Results and Discussion parts. Better to make them shorter, they must sound more stong without superfluous description.

A remark on Scheme 1 in page 7: put one curve arrow from the middle of the C=C bond to the carbocationic center instead of two curve arrows.

Author Response

Comments:
This paper describes a comparative study of four different acidic catalysts, such as, Amberlyst 15, heteropolyacids on graphite, sulfated zirconia, and NiO-Al2O3, on dimerization of isobutene into C8 olefins. The authors thoroughly studied a nature of acidic sites (Bronsted and Lewis) in these catalysts. They found that the the most effectively dimerization took place on catalysts possesing Bronsted acid sites. However, they stated that isobutene dimarization is not only surface process on acidic sites of a catalyst, but it can be very much dependable on textural properties of a catalyst. It was foubd that Amberlyst is the most effective catalyst for this dimerization.

In my opinion, the text of the paper is too long. Sometimes, it is difficult to follow the text, especially, when a lot of abbreviation are used in the text. My recomendation is to reduce Results and Discussion parts. Better to make them shorter, they must sound more stong without superfluous description.

A remark on Scheme 1 in page 7: put one curve arrow from the middle of the C=C bond to the carbocationic center instead of two curve arrows.

Response: The authors very much appreciate the general comment of the reviewer as well as the remarks concerning the paper. The length of the sections L174-L506 “Results and Discussion” have been revised and reduced to a great extent in the revised manuscript. On the other hand, Scheme 1 in page 7 has been modified according to the referee’s suggestions.

Reviewer 3 Report

The paper deals with determination of acidic sites over 4 catalysts used for IBE dimerisation. 

It contains some serious drawbacks, therefore it requires a major revision before being published.

My main comments are listed below:

Line 69 - the "Amberlyst" name should be linked to a certain catalyst mentioned in the previous sentence.

Line 73-74 - the NiO itself is not a reaction intermediate. It can be considered as a part of an intermediate formed after IBE adsorption. But what is the active site for IBE adsorption: NiO or Ni?

Line 75 - carbonation or carbocation?

The paper's drawback is the language. It must be corrected for English. Only some of the errors detected:

Line 102 - STA loading OF 15 wt%

Line 109 - helium flow OF 20 mL/min (and so on..)

Line 135 - He instead of he

Line 177 - the strength of acid sites instead of acidity strength

Please add an explanation, what was the aim of calculating the LT/HT ratio of acidic sites.

The information of the concentration of weak and stron acidic sites (given e.g. per 1 g of catalyst) is required to compare those materials with each other and to discuss their catalytic performance in IBE dimerisation.

Table 1 - the format of weight loss values is wrong

Line 237 - Table 1 instead of Table 2

The mass loss for A15 is significantly higher than for other thus its TGA profile should be displayed in Fig. 3.

The DTG and DTA would help to understand the character of mass loss at particular T ranges.

Line 255 - Fig 4 shows catalyst (not IBE) activity.

The unit of the catalytic activity on Y axis must be corrected (what is "μmol conv"?).

For discussing the deactivation, the TGA of spent catalysts is a must.

Round 2

Reviewer 3 Report

The Authors addressed some of my comments, but there are issues that still need to be corrected or discussed:

- wrong format of weight loss in table 1 (to be clear - 41.3% instead of 41'3%, etc.)

- Fig 3. It should be clearly indicated, which plots are TG and which are DTG. In the text, there is not a word about Fig. 3.

Moreover, from line 231 the Authors write: "The analysis of TPD-Ms showed the evolution of m/z 64 ascribed to desorption of SO2 on ZS, between 350-390 °C associated with sulfonic acids groups. While STA/HSAG100 and NiO/Al2O3 reveal a weight loss due to decomposition at higher temperatures (> 350 °C)".

What is the mass loss of the ZS sample between 350-390 °C (due to SO2 desorption)? 

What are the mass loses of STA/HSAG100 and NiO/Al2O3 at > 350°C?

The TGA part of the manuscript is actually a "space filler" because it does not provide any essential information regarding the catalysts performance in IBE conversion. The authors did not use the information they got from TGA-DTG.

I still think that discussion on catalyst deactivation needs to be evidenced with results of catalyst characterisation after tests. The decrease of its activity observed at the beginning of the reaction is one thing, but the reason for this loss of activity needs to be shown.

line 82: deactivation of the catalyst (instead of "reaction")

I recommend the revision of this manuscript according to the above comments.

Author Response

Reviewer #3:

Comments:

The Authors addressed some of my comments, but there are issues that still need to be corrected or discussed:

- wrong format of weight loss in table 1 (to be clear - 41.3% instead of 41'3%, etc.)

Response: We fully agree with the reviewer. This aspect has been rectified in the revised manuscript.

- Fig 3. It should be clearly indicated, which plots are TG and which are DTG. In the text, there is not a word about Fig. 3.

Response: We very much appreciate the critical evaluation of the reviewer, which has helped us to improve the manuscript. The changes implemented in the manuscript can be summarized as follows:

-We have modified the graph 3 accordingly.

-We have cited the figure 3 by twice, please refer to line 230 (Table 1 and Figure 3 summarizes the total weight loss at 600 °C for the studied catalysts) and line 238 (The corresponding TGA and DTG curves are presented in Figure 3) in the revised manuscript.

Moreover, from line 231 the Authors write: "The analysis of TPD-Ms showed the evolution of m/z 64 ascribed to desorption of SO2 on ZS, between 350-390 °C associated with sulfonic acids groups. While STA/HSAG100 and NiO/Al2O3 reveal a weight loss due to decomposition at higher temperatures (> 350 °C)".

What is the mass loss of the ZS sample between 350-390 °C (due to SO2 desorption)? 

Response: As it was already indicated in the original manuscript (please view line 236), m/z 64 is ascribed to desorption of SO2 on ZS, associated with sulfonic acid groups.

What are the mass loses of STA/HSAG100 and NiO/Al2O3 at > 350°C?

Response: The mass losses can be assigned to the decomposition of the heteropolyacid (STA) or the decomposition of the Ni precursor, among other species.

The TGA part of the manuscript is actually a "space filler" because it does not provide any essential information regarding the catalysts performance in IBE conversion. The authors did not use the information they got from TGA-DTG.

Response: The TGA/DTG method provides information on the changes in the solid samples’ composition, the stability of the samples with respect to the loss of water and their subsequent thermal decomposition. The authors have used this information in order to know the stability of the catalysts under reaction. It has been indicated in the revised manuscript (line 121-123: The measured weight loss gives information on the changes in the solid sample composition and thermal stability features. A derivative weight loss curve (DTG) can be used to determine the temperature at which the rates of weight changes are maximum).

I still think that discussion on catalyst deactivation needs to be evidenced with results of catalyst characterisation after tests. The decrease of its activity observed at the beginning of the reaction is one thing, but the reason for this loss of activity needs to be shown.

Response: We fully agree with the reviewer and more studies are in progress to evaluate the catalysts and their catalytic performances, as well as the deactivation aspects in this reaction (including stabilities, recyclability, etc.)

line 82: deactivation of the catalyst (instead of "reaction")

Response: According to suggested by the referee, deactivation of the reaction has been changed by deactivation of the catalyst.

I recommend the revision of this manuscript according to the above comments.

This manuscript is a resubmission of an earlier submission. The following is a list of the peer review reports and author responses from that submission.

Round 1

Reviewer 1 Report

"Comparative study of different acidic surface structures in solid catalysts applied for the isobutene dimerization reaction" written by José M. Fernández-Morales et al. dealt with a comparative investigation of an ion exchange resin (Amberlite 15), a carbon supported heteropolyacid (H4SiW12O40·nH2O), a sulfated zirconium oxide and a NiO/Al2O3 catalyst for the dimerization of isobutene. In detail, the characterization of the acidic properties (determined through NH3-TPD) of these different catalysts was conducted and correlated with their reactivity.

The paper is well structured and well organised but suffer from several aspects. 

Hereafter a list of suggestions that the authors should consider in order to improve the quality of the paper:

  1. First of all, in this manuscript, the catalysts were not well demonstrated to be "nano-scaled" systems. On the countrary, they look to be on a very bigger scale. Amberlite, sulfated zirconium oxide, γ-Al2O3 and graphite HSAG100 are all commercial catalysts/supports having dimension on the micron scale. Please, stress the  nano-structures issues in order to make this study more appropriate in being published in this journal.
  2. Are you sure that Amberlite A15 was the correct commercial name? Amberlyst should be instead reported into Alfa Aesar catalougue. In any case,  every time a commercial support/catalyst was mentioned along with the paper, please provide the respective commercial code for it.
  3. in order to better offer a more detailed discussion, characterization data on the catalysts used for this investigation should be more detailed: more focused and quantitative data should be reported. BET surfaces and acid sites (mmol) per gram of catalysts should be at least evaluated and distinguished as low and high acidity sites.
  4. As for the reactivity, the authors mentioned that the study was carried out in a range of temperature (50-180°C), but in the section 2.3 and the discussion 3.2, only data for 180°C were reported and discussed. Please amend.  
  5. Please, provide an appropriate number for each equation.
  6. Please check the definition for the TP1, TP2 and  C8s Selectivity equations. It looks like the authors want to focus their attention on selectivity calculated respect with the formation of dimers only. If this is the case, please specify it. In a more general vision, the overall selectivity should be a very useful data, which should be defined and calculated as IBEs converted into a specific product (TP1, TP2 or C8s) respect with total IBEs converted in all products (in dimers, trimers, and other carbonaceous products). In that case, the denominator of the abovementioned equations should consider the reacting factor of IBE, which is 2 for dimers, 3 for trimers and so on. Please report in any case also this value, which is more interesting.
  7. In order to improve the quality of the discussion on the reactivity of different catalysts, more quantitative data should be reported. For example a TON number could be really useful and could help in defining the productivity per g of catalyst.  
  8. L486-488. This concept is crucial for the quality of the manuscript. But actually it looks like it was only "qualitatively" discussed and not immediately perceived. Please improve this part. In order to do it, consider the suggestion N.3 and please report conversion of IBE and specific production of TP1, TP2 and C8s respect with the mmol of acid sites per gram of catalyst. Even distinction among strong and weak acid could be useful in defining reactivity and selectvity. Only in this way, it should be more immediate the effect of textural structure of the catalyst. 
  9. Conclusions should be improved. At the end of a comparison study, detailed suggestions (quantitative data) in terms of catalysts properties should be reported in order to define high conversion and selectivities. Considerations reported in L504-511 were not experimentally demonstrated and are too obvious to be reported.
  10. At the end, the authors should provide the real value related by this study: actually all the used catalysts were already tested and reported for this kind of reaction. Please stress the novelty of results reported in this work.

The authors should revise the paper since overally several stilistic points need to be checked. Herewith a list was reported as examples, but please, provide a general overview.

  • L137 "IBE:he" should be "IBE:He".
  • L139 "under he" should be "under He".
  • L242 revise "is include" 
  • L309 "an vital role" should be "a vital role".
  • L327 "structural complexity and chemical." should be "structural  and chemical complexity." 
  • L403 "(mainly PT2)" should be "(mainly TP2)"
  • L467 "could to manage" should be "could manage"
  • L480-484 Please revise, it was very badly written
  • L518: Funding: “This research received no external funding” is in disagreement with what is reported in L520: Acknowledgemnt

Reviewer 2 Report

The authors reported the comparative study of different acidic surface structures in solid catalysts applied for the isobutene dimerization reaction.

However, this paper is too long compared to its scientific content. Especially, introduction and discussion are too long. Conclusions are somewhat long. Furthermore, it is complex, confusing, and difficult to understand. I cannot understand what the authors stated. The results and discussion should be presented more clearly. In addition, Alfa-Aesar does not sell Amberlite 15. Is it Amberlyst 15? The authors do not see the reagent well.

Reviewer 3 Report

The manuscript entitled “Comparative study of different acidic surface structures in solid catalysts applied for the isobutene dimerization reaction” by Fernandez-Morales and co-workers establish a comparative evaluation of different catalysts towards the gas-phase isomerization of isobutene into C8 derivatives. The article is well executed, the results are partially promising given the selectivities achieved and pave the way for further improvement. As a major downside, the morpho-chemical characterization of the catalysts lack of sufficient analysis. The current manuscript seems more appropriate for a catalyst journal profile. I would recommend the authors to provide additional information to better shape the manuscript to this specific journal´s scope. Major revision is recommended prior to a final decision:

1) Please provide the reducing conditions to activate the Ni catalyst in the Experimental section.

2) Additional morphochemical characterization of the different materials must be provided. SEM, EDX, TEM, STEM, XPS, FT-IR. Those samples that more common, commercial type must be still further detailed for non-expert readers. Addition of schematic drawing of surface groups would be also interesting. Identification of nanostructured centers. Evaluation of the reducing treatment on the catalyst morphologies.

3) In figure 1, please provide the assignations inside the graphs for better identification.

4) The information supplied in Table 1 can be easily implemented in Figure 1. Figure 2 is not providing new information with respect to Figure 1. Can the authors justify its presence?

5) I would encourage the authors to provide the TG profiles as a potential new figure instead of Table 2.

6) How were the catalytic runs up to 24 hours operated? Please provide further details.

7) It would be very interesting that the authors included the claimed preliminary analyses to optimize the catalytic test conditions. That information is interesting itself and can be helpful for the audience, given the challenging character of this reaction.

8) Please, whenever is possible, provide further details on the side products detected, either in the form of bar graphs or supporting table.

9) Please provide additional details on regeneration and reutilization studies performed with these materials.